# The Aversive Response of the Locust *Locusta migratoria* to 3-Octanone, an Odorant from Fungal Pathogens, Is Mediated by a Chemosensory Protein

Xiao Xu [1], Long Zhang [1,2] and Xingbo Zhao [1,*]

1 College of Animal Science and Technology, West Campus, China Agricultural University, Beijing 100094, China; xuxiao_cau@yeah.net (X.X.); locust@cau.edu.cn (L.Z.)
2 Shandong Provincial Engineering Technology Research Center on Biocontrol for Pests, Jinan 250100, China
* Correspondence: zhxb@cau.edu.cn

**Abstract:** (1) Locusts are important agricultural pests. Identifying harmful substances and avoiding them is important for locusts' survival; their abilities to do so remain to be clarified. (2) We examined the electrophysiological (electroantennogram (EAG) and single sensillum recording (SSR)) and behavioral responses (preference behavior in a T-maze) of locusts to 18 different compounds; (3) Of these 18 compounds, 9 elicited strong EAG responses, and 3 elicited SSR responses of neurons expressing locust odorant receptor 3 (*Lmig*OR3). The 11 chemicals that elicited stronger EAG or SSR responses were selected for evaluation of the behavioral responses of locusts. Only 2-heptanone induced significant attraction responses in locusts at the tested concentration. RNA interference (RNAi) of *Lmig*OR3 and SSR experiments revealed that *Lmig*OR3 could detect 2-heptanone and 3-octanone. However, in behavioral experiments, RNAi of *Lmig*OR3 did not alter 2-heptanone-induced attraction but increased attraction by 3-octanone. (4) Our results suggest that the broadly tuned receptor expressed in a heterologous expression system exhibits a narrow electrophysiological response spectrum, and the aversive response of locusts to 3-octanone, an odorant from fungal pathogens, natural enemies, and non-host plants, is mediated by *Lmig*OR3. These findings enhance our understanding of the complex olfactory recognition mechanism in insects.

**Keywords:** locusts; electrophysiological responses; behavioral responses; *Lmig*OR3; 3-octanone; aversiveness

## 1. Introduction

The locust is an important agricultural pest worldwide. Their gregariousness, dispersal capacity, and voracious feeding habitats have had major negative effects on agricultural production, and they remain a major threat to various countries [1,2]. The largest recent outbreaks of the migratory locust, *Locusta migratoria*, have included two invasions in Madagascar from 1997 to 2000 (4.2 million hectares, US$ 50 million) and from 2013 to 2016 (2.3 million ha, US$ 37 million) [1] and invasions in Zambia (more than 10,000 ha, US$4 million) in 2017 [http://www.herald.co.zw/region-needs4m-to-fight-locusts/ (accessed on 18 July 2023)]. Continuous monitoring and control of locusts are needed given their potential to induce significant economic damage [3]. Some important behaviors of locusts are regulated by chemical perception [1,4], such as foraging [5], gathering [6], evasion of predators [7], and cannibalism [8].

Identifying and avoiding harmful substances in the environment is important for the survival of organisms that rely on chemical senses. Pathogenic microorganisms cause massive die-offs of susceptible insect populations [9], and insects avoid becoming infected themselves by recognizing and avoiding microbial emissions [10]. *Anthocoris nemorum* L. (Heteroptera: Anthocoridae) detects and avoids pathogenic *Beauveria bassiana* (Balsamo) Vuillemin (Ascomycetes: Hypocreales) when it forages on host plants [11]. The termite

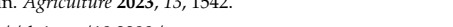

*Macrotermes michaelseni* has been reported to detect and avoid the pathogen *Metarhizium anisopliae* [12]. *Drosophila* can detect geosmin produced by various fungi [13], bacteria [14], and cyanobacteria [15] and then avoid these pathogenic microorganisms [16]. However, the olfactory mechanism by which locusts recognize these harmful substances is not well understood. The study of this mechanism is important for clarifying interactions between pathogenic microorganisms and locusts and might provide new insights that could be used to improve the efficacy of microbial pesticides.

Olfactory receptors (ORs), the main functional proteins of insect olfactory perception, play an important role in the recognition of complex chemical cues (odors) in the environment, olfactory coding, and mediating insect behavior; they are the key proteins in studies of olfactory-mediated behavior. Perception and signal transduction of external chemical signals by ORs are the initial biological and biochemical activities that occur in the peripheral nervous system, which involve peripheral nerve events [4,17]. The study of the tuning curve and dose-dependent response curve of compounds that induce the activation of ORs is the first step in understanding the olfactory coding of insects.

The "empty neuron" system in *Drosophila* can be used in non-model insects to study the chemical reaction spectrum of ORs; it accurately reflects the reaction characteristics of exogenous ORs and can be used to measure the specificity and sensitivity of OR reactions [18]. This has become the main system used for studying the functions of ORs [8,19–21]. Nevertheless, previous studies have found that the "empty neuron" system has certain limitations. For example, the ideal effect of the system was observed when mosquito ORs were expressed; however, the ideal effect was not observed in a system expressing moth ORs [20,22].

Locust OR3 (*Lmig*OR3) has been reported to be expressed in the trichoid sensilla, and its chemoreceptor spectrum has been obtained using the empty neuron system [23]. Recently, functional studies of many ORs of locusts have been conducted by expressing them in empty neurons of *Drosophila* [8,21,23].

Here, we examined the electroantennogram (EAG) and single sensillum recording (SSR) responses of locusts elicited by compounds that can induce pronounced electrophysiological responses of *Lmig*OR3 expressed in the empty neuron system. We found that the odorant receptor *Lmig*OR3 has a narrow odorant spectrum in locust, which contrasts with its spectrum in a heterologous expression system. We then characterized the behavioral responses of locusts to these compounds. We found that 3-octanone induces rejection behavior following its detection by *Lmig*OR3.

## 2. Materials and Methods

### 2.1. Animals

Locusts (*Locusta migratoria*: Orthoptera) were obtained from China Agricultural University. For the electroantennogram (EAG) experiment, the antennae of the fourth-instar nymph were used. Additionally, whole-mount fluorescence in situ hybridization and single sensillum recording (SSR) experiments also used the antennae of the fourth-instar nymph. For the preference behavior bioassay, fourth-instar nymphs were used. The procedures for rearing and tissue extraction from antennae of fourth-instar nymph were described in Xu et al. (2013) [24].

### 2.2. Electroantennogram (EAG)

The antennal receptivity of the fourth-instar nymphs to 18 selected chemicals was determined by EAG tests. The antennae were removed from the base, and the tip of the antenna was removed. Each dissected antenna was immediately fastened with electrode gel (Spectra 360 Electrode Gel) on two metal electrodes [25]. EAG tests were conducted at a temperature of $30 \pm 2\ °C$ and using incandescent light.

Each chemical tested was diluted in mineral oil to 1, 10, and 100 µg/µL for use in comparative EAG response tests. In brief, 1, 10, and 100 mg of compounds were dissolved in 1 mL of sodium mineral oil, respectively. To prepare treatments for testing, 10 µL of each

chemical solution was applied to a piece of filter paper strip (approximately $0.5 \times 2$ cm$^2$); the paper strip was then placed inside a glass Pasteur pipet (10 cm in length). The tip of the pipet was inserted into a small hole through the wall of a glass tube (0.2 cm diameter, 14 cm long) directed at the antennal preparation. A continuous flow rate (150 mL/min) of charcoal-filtered and humidified air was provided by a stimulus controller (CS-05, Syntech, Germany), with a stimulus duration of 1 s. A 60-s interval between successive stimulations was allowed for antennal recovery. The EAG responses to 10 μL of mineral oil were tested as a control; EAG responses to 10 μL of hexanal were tested and used as a reference stimulus. Responses to blank (mineral oil) and standard (hexanal) treatments were obtained before and after all the test chemicals so that corrections could be made in the event of a loss of sensitivity of the preparation during the recording. The decrease in sensitivity was assumed to be linear with time. EAG recordings were obtained from 11 to 12 antennae for each solution. Signals were stored and analyzed using Syntech EAG 2000 (the Netherlands).

A total of 18 chemicals were purchased from Sigma-Aldrich Company (Shanghai, China); the highest grade of the chemicals available (97–99%) was purchased (Table S1). The chemicals were then diluted to different concentrations and used as stimulants with mineral oil.

### 2.3. Whole-Mount Fluorescence In Situ Hybridization (WM-FISH)

Templates of *Lmig*OR3 were generated by standard PCR using gene-specific primer pairs (Table S2, *Lmig*OR3-probe-s/as (WM-FISH)). Digoxigenin (DIG)-labeled antisense probes were generated from linearized recombinant pGem-T Easy plasmids using the T7/SP6 RNA transcription system (Roche, Basel, Switzerland) following recommended protocols.

A previously described protocol for RNA WM-FISH was used with modifications [26,27]. Briefly, the antennae were dissected from fourth-instar locust nymphs and transferred to a series of fixing and washing solutions at different times (Table S3). Next, pre-hybridization was conducted at 55 °C for at least 6 h with an in situ hybridization solution (pH 8.0, Table S3). The antennae were incubated in a hybridization solution containing labeled antisense RNA probes (1:100) at 55 °C for at least 48 h. After hybridization, the antennae were blocked with 1% blocking reagent (Roche, Germany), incubated with an anti-digoxigenin alkaline phosphatase-conjugated antibody (Roche, Germany), visualized with HNPP, and made into slides; a series of washes were interspersed between the above steps and the detailed procedures are provided in Schultze et al. (2013) and Xu et al. (2017) [26,27]. Images were captured using an Olympus BX45 confocal microscope and analyzed using FV1000 software (4.2b). Images were not altered except for the uniform adjustment of brightness or contrast within a single figure. The number of samples observed was 3 ($n = 3$), which meant that the presence of the same signal was observed on three different slides.

### 2.4. Single Sensillum Recording (SSR)

SSR was conducted using a $10\times$ universal AC/DC amplifier (Syntech, The Netherlands); the signals were recorded in an Intelligent Data Acquisition Controller (IDAC-4, Syntech, The Netherlands) and viewed on a personal computer. AutoSpike32 software (3.7.0.0) was used to record the spikes after stimulation.

SSR and antennal preparation procedures of fourth-instar females of *L. migratoria* were performed following Cui et al. (2011) and Li et al. (2018) [28,29]. The locust was placed in a plastic tube (1 cm in diameter), and its antennae were fixed on a glass plate with double-sided adhesive tape. Tungsten wire electrodes were sharpened electrolytically with 10% NaNO$_2$. The recording electrode was inserted in the base of the sensillum, and the reference electrode (tungsten electrode) was inserted into the head of the locust using a micromanipulator (Narishige, Japan).

Diluted volatile compounds (each 10 μL) were applied to folded strips of filter paper (length 2 cm, width 0.5 cm), which were inserted into Pasteur tubes. Each Pasteur tube was only used for testing a specific compound. Mineral oil was used as a blank control. The

tube carried a constant airflow (150 mL/min), and its opening was positioned 1 cm from the antenna. The odor airflow was controlled by a stimulus air controller (CS-55, Syntech, The Netherlands) and directed to the surface of the antenna. The stimuli were provided over 1 s. The recovery period was 1 min. The number of samples tested was 6–7.

### 2.5. Preference Behavioral Bioassay

Bioassays were performed in a glass two-choice olfactometer following Obeng-Ofori et al. (1993) [30] under uniform illumination and a temperature of 30 ± 2 °C. Diluted volatile compounds (each 10 µL) were applied to folded strips of filter paper (length 2 cm, width 1 cm), which were placed into one glass arm as a stimulus odor source; the other glass arm was placed with a filter paper coated with mineral oil (10 µL) as a control.

Each replicate contained 20 locusts, and 3–5 replicates of all tests were performed. A fourth-instar nymph was released individually into the olfactometer to observe its behavior, and each individual was only used in tests once. The preference of each locust was observed and recorded within 10 min. When the locust entered one side of the glass arm with its whole body and stayed for 30 s, it was considered to have made a choice; otherwise, it was considered to have made no choice. If the locust did not make a choice, it was discarded and did not participate in the calculation of the preference index. The preference index (PI) was calculated as (T-C)/N, where T is the number of locusts in the odor source arm, C is the number of locusts in the control arm, and N is the total number of tested locusts.

### 2.6. RNA Interference (RNAi)

The target double-stranded RNA (dsRNA) was synthesized per the manufacturer's instructions. PCR products were amplified with the T7 promoter-conjugated primer; primer pairs are shown in Supplementary Table S1. The PCR thermal cycling parameters were as follows: initial denaturation at 94 °C for 5 min; 35 cycles of 94 °C for 30 s, 60 °C for 30 s, and 72 °C for 1 min; and a final extension at 72 °C for 10 min. The dsRNA was synthesized using a T7 RiboMAX™ Express RNAi System (Promega, the United States), diluted to 2 µg/mL with distilled deionized water, and stored at −20 °C. The detailed procedures for dsRNA synthesis are provided by Li et al. (2018) [29].

Target dsRNA (10 µg) was inserted into the dorsal vessel of each locust using a microinjector, and dsGFP was used as a control. The efficiency of RNA silencing was evaluated on the third day after injections.

### 2.7. Real-Time Quantitative PCR (qPCR)

The locusts' antennae were placed in Eppendorf tubes chilled on liquid nitrogen and homogenized with ceramic beads for 180 s at 60 Hz in a tissue lyzer. Total RNA was isolated using the TRIzol reagent (Invitrogen, Waltham, MA, USA) following the manufacturer's protocol. cDNA was synthesized from 1 µg of total RNA using the GoScript™ reverse transcription system (Promega, Madison, WI, USA) for qPCR. We designed qPCR primers to amplify 100–250-bp products from the unigene sequences (Supplementary Table S2). The primers were assessed using normal PCR (TaKaRa, Dalian, Liaoning, China) and sequencing to verify that the products were correct and that there were no primer dimers. The $2^{-\Delta\Delta CT}$ method was used to quantify the relative expression levels of each gene. The expression levels of the genes were normalized against those of the reference gene *LmigActin*. qPCR was conducted in 20 µL reactions (including 10 µL of SuperReal PreMix, Tiangen, Beijing, China) on an ABI QuantStudio 6 Flex qPCR platform (USA) with the following PCR program: 95 °C for 15 min, and 40 cycles of 95 °C for 10 s and 60 °C for 30 s. Melting curve analysis was conducted by increasing the temperature from 60 °C to 95 °C and evaluating the specificity of the real-time PCR products. Three technical replicates were performed for each sample.

*2.8. Statistical Analysis*

The results of behavioral experiments were analyzed using one-way ANOVA followed by Fisher's LSD test to evaluate the difference in the preference index between tested compounds and CK. A one-way ANOVA followed by Dunnett's multiple comparison tests was used to evaluate the difference between tested compounds and CK in EAG results. SSR results after RNAi were analyzed using one-way ANOVA followed by Tukey's multiple comparison tests to assess the difference in SSR between wildtype (WT), dsGFP, and dsOR3. A one-way ANOVA followed by Fisher's LSD test was used to evaluate the difference in expression level between WT, dsGFP, and dsOR3 in qPCR results. All figures were made using GraphPad Prism 6 (GraphPad Prism 6.0.1 Software, San Diego, CA, USA).

## 3. Results

*3.1. EAG Responses of Locusts to 18 Compounds at Different Concentrations*

The 18 compounds tested included 7 esters with C5–C8 carbons, 5 heterocyclic compounds with C5–C9 carbons, 5 ketones with C7–C9 carbons, and 1 benzyl alcohol. Before and after each test of the EAG response to 18 compounds, 10% hexanal was used as a positive control to detect whether the isolated antennae were still active. The results showed that there was no significant difference in the EAG responses induced by 10% hexanal before and after the experiment (Figure S1A), indicating that antennae were active throughout the experiment process. The EAG responses to different compounds at the same concentration were compared, and the results showed that the EAG responses of locust antennae to 18 tested chemicals did not significantly differ from those of the control (mineral oil, CK) at a concentration of 1 μg/μL (Figure 1A). Five compounds (benzyl alcohol, 4, 5-dimethylthiazole, hexyl acetate, 2-octanone, and 2-heptanone) evoked EAG active responses of locusts at a concentration of 10 μg/μL (Figure 1B). The strongest EAG response was to benzyl alcohol ($0.207 \pm 0.119$ mV), followed by 4, 5-dimethylthiazole ($0.196 \pm 0.080$ mV), and hexyl acetate ($0.148 \pm 0.123$ mV). Except for benzyl alcohol, the EAG reaction intensity increased when locusts were stimulated with 17 compounds at a concentration of 100 μg/μL, but the magnitude of the increase varied among the compounds. The EAG responses evoked by pentyl acetate, butyl acetate, 3-heptanone, 2, 5-dimethypyrazine, and 2, 4, 5-trimethylthiazole was stronger compared with the reactions induced by the other compounds, and significant differences from the control were observed for the reactions evoked by these compounds; these compounds replaced benzyl alcohol as the compounds inducing the active EAG reactions at high concentrations (Figure 1C). Among all 18 compounds, there were four main groups (ordered from strongest to weakest responses):

1.    2-heptanone, 4, 5-dimethylthiazole, and 2, 4, 5-trimethylthiazole;
2.    2, 5-dimethypyrazine;
3.    3-heptanone, 2-octanone, and penthyl acetate;
4.    3-octanone and butyl acetate (Figure 1C).

A comparison of the EAG responses induced by the same compound at different concentrations revealed that the intensity of the responses induced by the 16 compounds increased significantly as the concentration applied increased; however, the magnitude of the increase varied (Figures S1B and S2). The rate of increase was highest for 2, 5-dimethylpyrazine, but the EAG reaction was low when its concentration was low, and an EAG active reaction was only observed at a concentration of 100 μg/μL (Figure S1B). The rate of increase in EAG of 2-heptanone, 4, 5-dimethylthiazole, and 2-octanone was not high, but strong reactions were induced at all concentrations from 10 to 100 μg/μL (Figure 1A–F), indicating that locusts show broad sensitivities to these three compounds. Conversely, benzyl alcohol was the only compound among tested substances where the EAG response intensity reached saturation and decreased instead of increasing at a concentration of 100 μg/μL (Figure 1G).

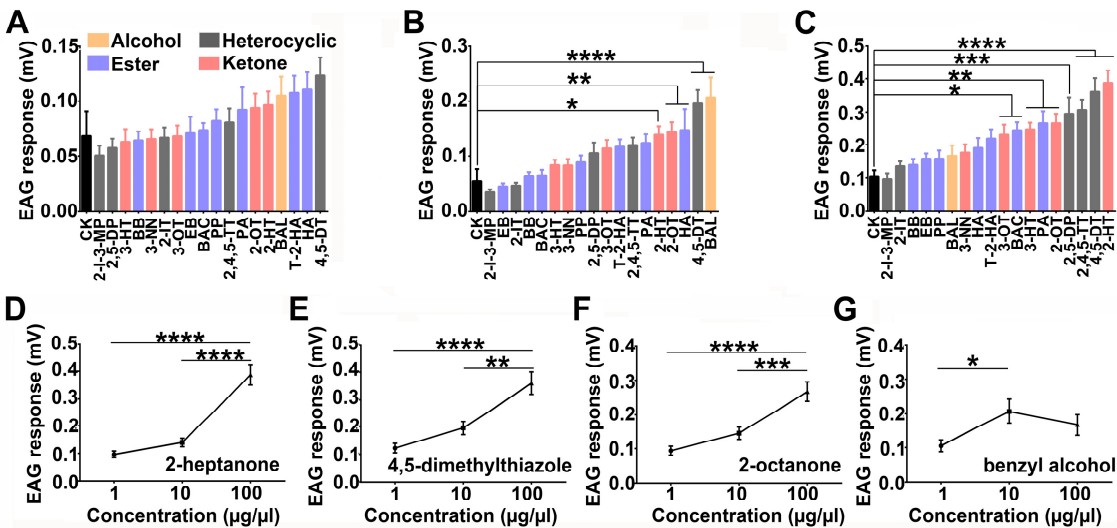

**Figure 1.** The EAG responses of locust antennae to 18 chemicals. EAG values and bars are mean ± SEM (mV). $n$ = 11–12. Stimulus dose were with 10 μL of 1 μg/μL (**A**), 10 μg/μL (**B**) and 100 μg/μL (**C**). Univariate analysis of variance (ANOVA) and Dunnett's multiple comparison tests ($α$ = 0.05) were used to compare the difference in EAG responses of the same dose of different odor compounds and CK. CK is mineral oil (solvent) as odor stimulus. 2-I-3-MP: 2-isobutyl-3-methoxy-pyrazine; 2, 5-DP: 2, 5-dimethylpyrazine; 3-HT: 3-heptanone; BB: butyl butyrate; 3-NN: 3-nonanone; 2-IT: 2-isobutylthiazole; 3-OT: 3-octanone; EB: ethyl butyrate; BAC: butyl acetate; PP: pentyl propionate; 2, 4, 5-TT: 2, 4, 5-trimethyl thiazole; PA: pentyl acetate; 2-OT: 2-octanone; 2-HT: 2-heptanone; BAL: benzyl alcohol; T-2-HA: trans-2-hexenyl acetate; HA: hexyl acetate; 4, 5-DT: 4, 5-dimethylthiazole. *, $p < 0.05$. **, $p < 0.01$. ***, $p < 0.001$. ****, $p < 0.0001$. (**D–G**): EAG responses of locust antennae to different concentrations of 2-heptanone, 4, 5-dimethylthiazole, 2-octanone, and benzyl alcohol. $n$ = 11–12. Mean EAG responses to the applied doses of the same odor compound were compared by ANOVA and Tukey's HSD test (mean EAG response <0.05. **, $p < 0.01$. ***, $p < 0.001$. ****, $p < 0.0001$.

### 3.2. Responses of Neurons in One Trichoid Sensilla to 18 Compounds

To understand the responses of neurons in trichoid sensilla to these 18 compounds, we first performed in situ hybridization experiments with the locust odorant receptor *Lmig*OR3 and found that *Lmig*OR3 was expressed in neurons housed in trichoid sensilla (Figure 2A). In these trichoid sensilla, neurons A and B were identified according to the amplitude of spontaneous spikes (Figure 2B). An active response evoked by a stimulant was defined to be |Δspikes| greater than twice the mean standard deviation of spikes caused by a control group (PO) [31], and if the emission frequency was greater than 5 spikes per second of the response compared with the control (|Δspikes| > 5), the response of the neuron was excited or inhibited.

When stimulated at a concentration of 1 μg/μL, none of the 18 compounds could trigger a response by neuron A, and two compounds (pentyl acetate and 2, 4, 5-trimethyl thiazole) induced an inhibitory response by neuron B (Figure 2C). At a dose of 10 μg/μL, eight compounds enhanced the inhibitory response to neuron B; the strongest reaction was induced by pentyl acetate (Δspikes = −11.17 ± 9.03 spikes/s), followed by butyl acetate (Δspikes = −8.33 ± 9.36 spikes/s). By contrast, all 18 compounds still could not trigger a response by neuron A. When the dose of the compound was increased to 100 μg/μL, 16 compounds elicited an inhibitory response by neuron B, with an exception of 2-isobutyl-3-methoxy-pyrazine and ethyl butyrate. Furthermore, seven chemicals induced strong responses (|Δspikes| > 20 spikes/s, 20 spikes/s is approximately four times the mean standard deviation of the firing frequencies of neuron A compared with the control stimuli). Conversely, three compounds could stimulate the excitatory response of neuron A (2, 5-dimethylpyrazine, 2-octanone, and 2-heptanone), and the strongest SSR response was

induced by 2, 5-dimethylpyrazine ($\Delta$spikes = 7.39 $\pm$ 5.75 spikes/s), followed by 2-octanone ($\Delta$spikes = 6 $\pm$ 6.09 spikes/s) and 2-heptanone ($\Delta$spikes = 5.05 $\pm$ 5.09 spikes/s).

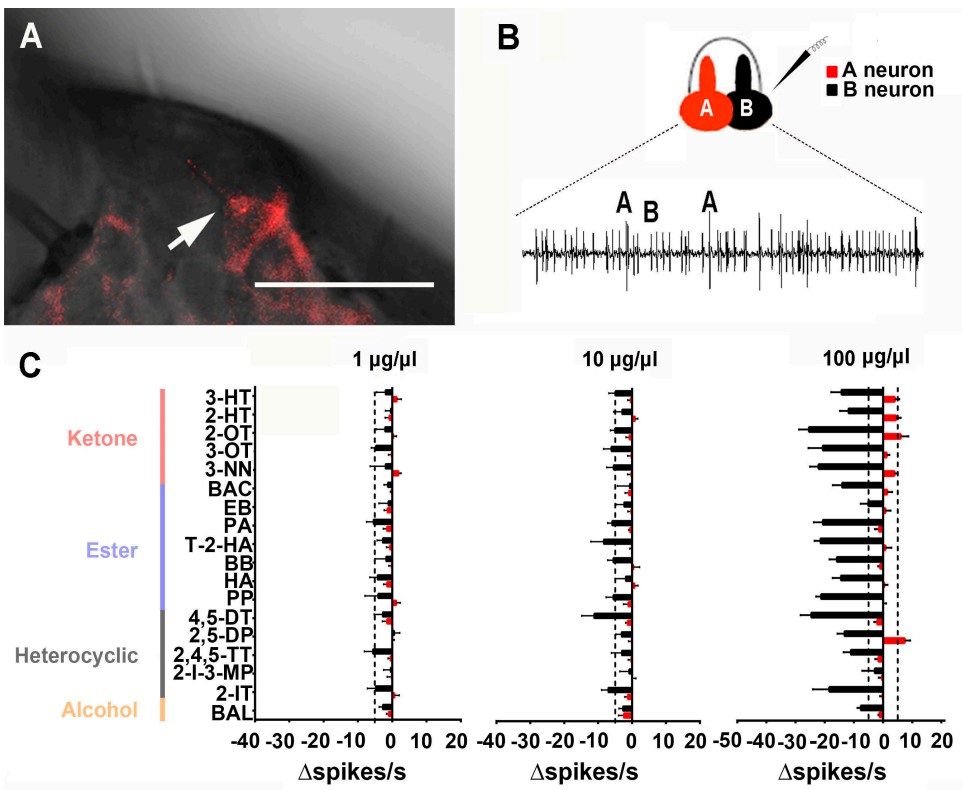

**Figure 2.** SSR responses of one trichoid sensilla to 18 compounds. (**A**): In situ fluorescence hybridization revealing that *Lmig*OR3 was expressed in olfactory sensory neurons housed in trichoid sensilla. White arrows indicate trichoid sensilla expressing *Lmig*OR3, and the white scale bars represent 40 μm. (**B**): Representative spontaneous potential spikes from SSR recordings from locust trichoid sensilla. One sensillum type and two ORN classes of the antenna are also shown. (**C**): SSR response of one trichoid sensillum that expressed *Lmig*OR3 to 10 μL of 18 chemicals at different concentrations. *n* = 6–11. 2-I-3-MP: 2-isobutyl-3-methoxy-pyrazine; 2, 5-DP: 2, 5-dimethylpyrazine; 3-HT: 3-heptanone; BB: butyl butyrate; 3-NN: 3-nonanone; 2-IT: 2-isobutylthiazole; 3-OT: 3-octanone; EB: ethyl butyrate; BAC: butyl acetate; PP: pentyl propionate; 2, 4, 5-TT: 2, 4, 5-trimethyl thiazole; PA: pentyl acetate; 2-OT: 2-octanone; 2-HT: 2-heptanone; BAL: benzyl alcohol; T-2-HA: trans-2-hexenyl acetate; HA: hexyl acetate; 4, 5-DT: 4, 5-dimethylthiazole. The dashed line represents |$\Delta$ the spike| = 5, and SSR responses larger than the dashed line were active responses as defined in this paper.

### 3.3. Preference Behavior of Locusts in Response to 11 Compounds

According to the results of EAG and SSR experiments, the preference behavior of the locusts was examined for 11 compounds. As the distance from the source of the odor to the entrance of insect (the length of the single arm of the T-maze device was about 49 cm) would weaken the odor concentration practically perceived by locusts [32], and all compounds just induced low EAG responses at 1 μg/μL, we excluded 1 μg/μL here and only compared 10 μg/μL and 100 μg/μL to make sure the concentration was high enough to be perceived by the locusts. The concentration of 10 μg/μL was closer to that observed in the natural habitat of locusts compared with 100 μg/μL [21]. Therefore, for those (4, 5-dimethylthiazole, benzyl alcohol, hexyl acetate, and trans-2-hexenyl acetate) that could induce strong EAG responses at 10 μg/μL, a concentration of 10 μg/μL was used in subsequent behavioral experiments, and 100 μg/μL was selected for the rest. The

results showed that 2-heptanone, a volatile substance derived from the main food source of locusts, significantly attracted locusts (Figure 3).

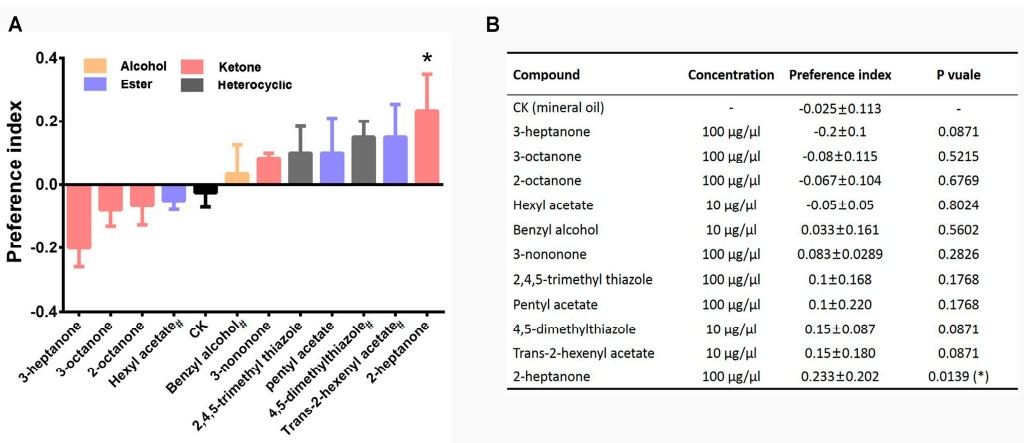

**Figure 3.** Behavioral responses of locusts to 11 chemicals in olfactometer bioassays (**A**,**B**). (**A**): The "#" marked on the lower right corner of the compound indicates that its concentration was 10 μg/μL in behavioral experiments. No marker indicates that the concentration was 100 μg/μL. (**A**,**B**): The preference index (PI) was calculated as (T-C)/N, where T is the number of locusts in the odor source arm, C is the number of locusts in the control arm, and N is the total number of tested locusts. Each test was within 10 min. CK is mineral oil (solvent) as an odor stimulus. The results of behavioral experiments were analyzed using one-way ANOVA and Fisher's LSD test. A single asterisk (*) indicates a significant difference at $p < 0.05$. Bar is S.E.M. $n$ = 3–5 (20 locusts per replicate) for each independent test.

### 3.4. LmigOR3 Is Expressed in Neuron A in the Tested Sensilla and Mediates 3-Octanone-Induced Rejection

We aimed to determine which neuron in the trichoid sensilla expresses OR3 and whether OR3 contributes to locust behavior induced by these 18 compounds. We reduced *Lmig*OR3 expression levels using RNAi (Figure 4A,B). Through SSR recordings, we showed that the response to 2-heptanone and 3-octanone was significantly reduced in neuron A of dsOR3 locusts compared with that of wild-type (WT) nymphs and dsGFP locusts, and the response of neuron B remained unaffected (Figure 4C,D), which indicated that *Lmig*OR3 is expressed in neuron A. Moreover, we tested the behavioral response of dsOR3 locusts to 2-heptanone and 3-octanone in the dual-choice olfactometer; we found that the dsOR3 locusts had completely lost their aversion and were attracted by 3-octanone compared with WT and dsGFP locusts, and no changes in behavior in response to 2-heptanone were observed (Figure 4E,F). We conclude that olfactory sensory neurons that express *Lmig*OR3 are present in neuron A of the tested trichoid sensillum and are responsible for the negative behavioral response to 3-octanone in locusts.

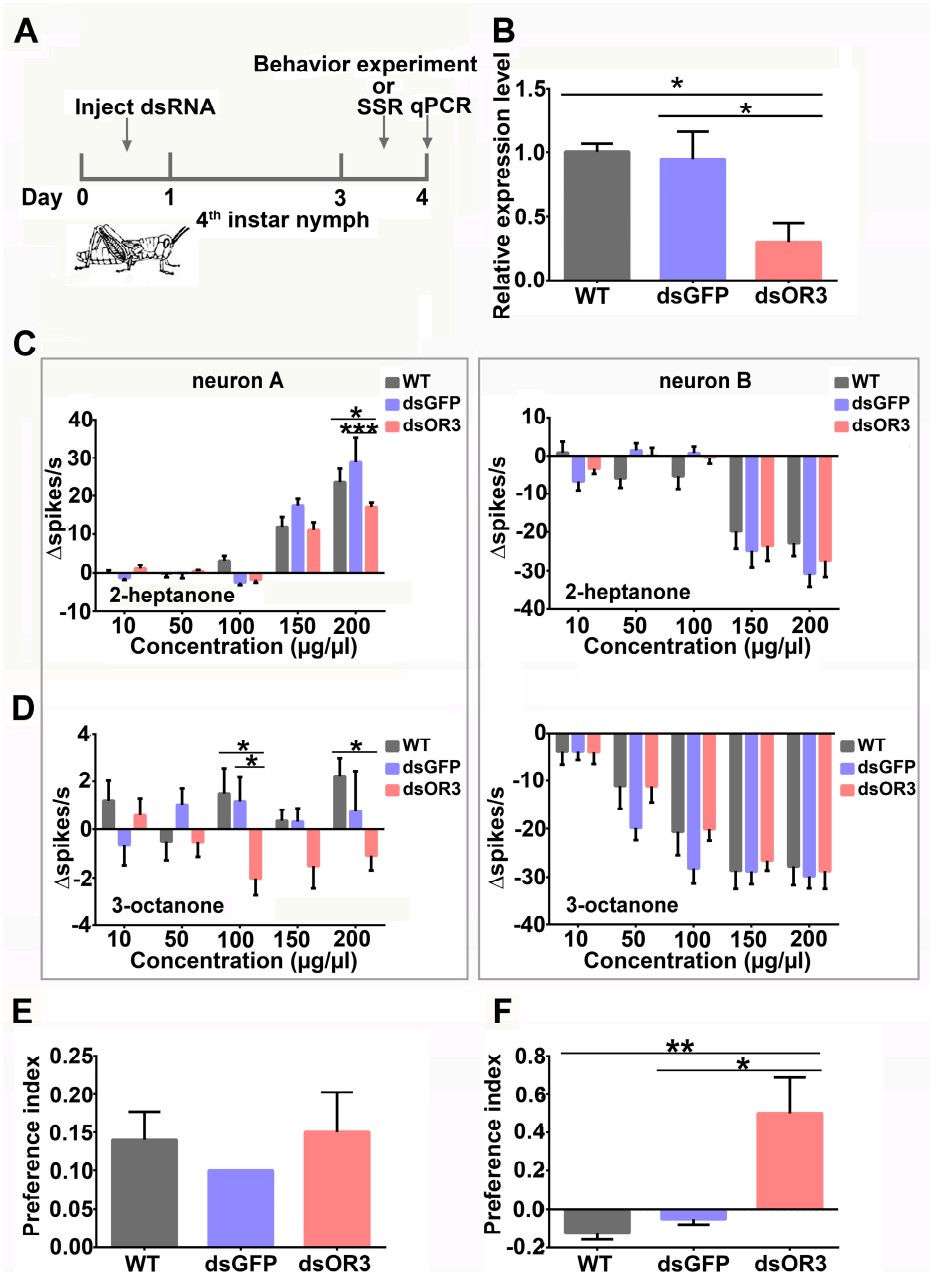

**Figure 4.** *Lmig*OR3 detects 3-octanone and 2-heptanone; it also mediates aversiveness to 3-octanone in locusts. (**A**): Timeline of the experiment from dsRNA injection (day 0–1 after the 4th ecdysis), behavioral or SSR experiments (days 3–4), and detection of the dsOR3 interference effect (qPCR on days 4). (**B**): qPCR experiment showing the effects of injection of OR3 and GFP dsRNA. $n = 3$. Ordinary one-way ANOVA. Fisher's multiple comparisons test. Error bar: S.E.M. *, $p < 0.05$. (**C**,**D**): Comparison of the number of neurons A and B spikes housed on trichoid sensillum expressing *Lmig*OR3 in WT, dsGFP, and dsOR3 locusts in response to 2-heptanone and 3-octanone at different dosages. Ordinary two-way ANOVA. Tukey's multiple comparisons test. *, $p < 0.05$; ***, $p < 0.001$; Error bar: S.E.M. $n = 5–9$. (**E**): Behavioral responses of WT, dsGFP, and dsOR3 locusts to 2-heptanone. (**F**): Behavioral responses of WT, dsGFP, and dsOR3 locusts to 3-octanone. Ordinary one-way ANOVA. Tukey's multiple comparisons test. *, $p < 0.05$; **, $p < 0.01$. Error bar: S.E.M. $n = 3–5$.

## 4. Discussion

The response spectra of *Lmig*OR3 to these 18 compounds in locusts and "at1 empty neurons" in *Drosophila* were compared. In contrast to the strong excitatory responses in transgenic *Drosophila* [23], the odorant response profile became narrower when *Lmig*OR3

was expressed in locusts (Figure 2). Among the 18 compounds, only 3 compounds (2, 5-dimethylpyrazine, 3-heptanone, and 2-heptanone) could induce excitatory reactions; most of the others did not respond after stimulation. This result was in contrast to the performance of mosquito OR8 (AgOR8), which has a similar odorant response profile in both endogenous and exogenous neurons (only slightly narrower in the exogenous system than in the endogenous system; the compounds induced excitatory responses from 12 to 9) [20]. Given that these 18 compounds can stimulate strong inhibitory responses of B neurons in locust, the change in the response spectrum of *Lmig*OR3 might stem from the competitive binding of different ORs to the same compound, or the interaction between different neurons in the same sensillum [19]. Furthermore, the differential sensitivity to these ligands observed between at1 empty neurons expressing *Lmig*OR3 and endogenous neurons expressing *Lmig*OR3 differed; specifically, the intensity of compounds that can trigger excitatory reactions decreased from 30 spikes/s $< n$ (exogenous neuron) to $5 < n < 10$ spikes/s (endogenous neuron) [23], and some compounds elicited responses that appeared excitatory in empty neurons but inhibitory in endogenous neurons, which was similar to AgOR8 [20]. This might stem from differences in the lymphatic environment and competitive combinations of other *Lmig*ORs [22]. Given that few concentrations were tested in this experiment and the gaps between the tested concentrations were large, we are unable to rule out the possibility that we did not use concentrations appropriate for activating neurons if the range of concentrations of these compounds capable of activating neurons is narrow. That is, our tested concentrations might not have been able to induce the response of A neurons. In conclusion, the expression of some locust ORs in allogeneic expression systems might not accurately reflect their function and response spectra.

Most compounds can induce an inhibitory response in neuron B. After reducing the stimulus concentration to 10 µg/µL and 1 µg/µL, the inhibitory reaction intensity induced by these 18 compounds weakened, and some compounds were no longer able to provoke an inhibitory reaction. These results indicate that the inhibitory response of neuron B induced by 18 compounds at 100 µg/µL might be mainly related to the type of compounds rather than the use of excessively high concentrations. However, given that only 18 compounds were tested in this experiment, whether B neurons can only be activated by inhibition remains unclear.

In the EAG experiment, 3-octanone did not provoke a response at 10 µg/µL, which was the concentration that was most used in the other experiments. The reason might be that there were some interactions between the neurons, in which some exhibited excitation and some exhibited inhibition responses to the chemical, and together they showed a lower EAG response [33,34].

Additionally, in the SSR experiments, 3-octanone only caused slight increases in the spikes of A neurons (Δspikes = 1.49 ± 2.82 spikes/s, Figure 2C), and the response did not meet the criteria for A neuron activation as defined in this paper. However, when the level of OR3 expression was reduced, the response of A neurons to 3-octanone shifted from excitatory to inhibitory (Δspikes = -2.03 ± 1.72 spikes/s, Figure 4D), and the response of locusts to 3-octanone changed from repulsion to attraction (Figure 4F). This suggests that *Lmig*OR3 expressed on A neurons can sense 3-octanone. Other 3-octanone-sensing ORs might induce inhibitory responses to 3-octanone in A neurons when activated. Neurons that can be bidirectionally activated in *Drosophila* increase the odor coding ability by decreasing response saturation and can drive two opposite behaviors, with a shift in neuronal response type leading to a complete switch in olfactory behavior [33], and A neurons might function similarly in locusts. The response of one ORN can be inhibited by the activation of a neighboring ORN, which can modulate olfactory behavior in *Drosophlia*; thus, the activation of B neurons by 3-octanone might inhibit the response of A neurons. Moreover, the response of A neurons induced by 3-octanone might be weak; however, the nonlinear signal amplification mechanism between olfactory neurons (ORN) and projector neurons (PNs) might increase the strength of the signal in PNs and finally induce avoidance

behavior in locusts [35,36]. Additional experiments are needed to confirm the above hypothesis, given that this was not examined in our study.

The increase in spikes in A neurons induced by 3-heptanone and 3-nonanone was greater than that induced by 3-octanone, but the responses of the A neurons of dsOR3 and WT locusts to 3-heptanone and 3-nonanone did not significantly differ after the level of OR3 expression decreased (Figures 4D and S3A,B). There was also no significant change in the response of B neurons in dsOR3 locusts to 3-heptanone and 3-nonanone. Staining results have shown that the neurons of a mosquito can express multiple ORs at the same time [37]. Non-canonical coexpression in mosquitoes differed from that observed in *Drosophila*, which increases the robustness of the mosquito olfactory system by increasing its redundancy [37]. Changes in the SSR responses observed in our study suggested that the co-expression in locusts was possibly the same as that observed in mosquitoes; other ORs were activated by 3-heptanone and 3-nonanone in addition to OR3 in A neurons.

3-octanone is a volatile of *Metarhizium anisopliae* [38], *Isaria fumosorosea* K3 [39], and other insect pathogens; it has repellent and toxic effects on nematodes [40], *Galleria mellonella*, and other insects [41]. Furthermore, 3-octanone is an alarm pheromone in ants, the natural enemy of locusts [42], and a volatile in non-host plants such as mushrooms [43]. We did not investigate the toxicity of 3-octanone to locusts, but it is undoubtedly distasteful for locusts (Figure 4, preference index < 0), which might stem from their avoidance of dangerous things or because they find non-host plants distasteful. By contrast, 2-heptanone is a volatile in their host plant [5], and migratory locusts are highly sensitive to this volatile (Figures 1 and 3). Previous research has indicated that *Lmig*OR3 can sense 2-heptanone, but it is not involved in mediating 2-heptanone-induced attraction behavior. Other ORs might be responsible for mediating 2-heptanone-induced locust attraction behavior. In addition, 4, 5-dimethylthiazole attracts locusts, and 2-octanone repels locusts (Figure 3). We did not investigate whether OR3 senses these compounds and mediates induced behaviors. We might explore this possibility in future studies.

The results of this paper confirm that locusts can detect and avoid 3-octanone (a chemical derived from pathogens, natural enemies, and non-host plants), which is detected by *Lmig*OR3, and this fills a gap in our knowledge of the ability of locusts to identify harmful substances. 3-octanone could be used as a component of locust repellent to protect crops, and dsOR3 could be applied with microbial pesticides to weaken the ability of locusts to identify pathogenic microorganisms and increase locust morbidity. The feasibility of the above strategies requires experimental verification; however, the results of our study provide new insights that could aid future improvements in the control of locusts.

**Supplementary Materials:** The following supporting information can be downloaded at: https://www.mdpi.com/article/10.3390/agriculture13081542/s1, Figure S1: The EAG response of antennae; Figure S2: EAG responses of locust antennae to different concentrations of 14 compounds; Figure S3: Comparison of the number of neuron A and B spikes housed in trichoid sensillum in WT, dsGFP, and dsOR3 locusts in response to 3-heptanone (B1 and B2) and 3-nonanone (C1 and C2) at different dosages after RNAi (A); Table S1: The traits of all chemicals used; Table S2: The primers in experiments; Table S3: The solutions in whole-mount in situ hybridization.

**Author Contributions:** L.Z. and X.Z. proposed the idea; X.X. and L.Z. designed the experiments; X.X. conducted the experiments; X.X., L.Z. and X.Z. wrote the paper. All authors have read and agreed to the published version of the manuscript.

**Funding:** This work was supported by the National Nature Science Foundation of China [grant numbers 31872968]; and the Agricultural scientific and technological innovation project of Shandong Academy of Agricultural Sciences [CXGC2022E04].

**Institutional Review Board Statement:** Not applicable.

**Data Availability Statement:** Where no new data were created.

**Acknowledgments:** We thank the native language expert for editing the language of a draft of this manuscript.

**Conflicts of Interest:** The authors declare that the research was conducted in the absence of any commercial or financial relationships that could be construed as a potential conflict of interest.

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
