# Peer review of "The Aversive Response of the Locust Locusta migratoria to 3-Octanone, an Odorant from Fungal Pathogens, Is Mediated by a Chemosensory Protein"

_agriculture, doi:10.3390/agriculture13081542_

Round 1
Reviewer 1 Report
The Ms by Xu et al examined the electrophysiological and behavioral responses of locusts to different potential chemical cues. The authors tested the response of 4th instar antennae using bioassays with electroantennogram (EAG), single sensillum recording (SSR), in situ hybridization (WM-FISH), and olfactometer. They concluded LmigOR3 is involved in the detection of 3-octanone, a chemical that mediates aversive responses in locusts.
Although of some merit, the present Ms lack novelty considering previous similar work with the same topic (see: You et al., 2016; https://doi.org/10.1016/j.ibmb.2016.10.008). Furthermore, the Ms in its present form suffers many critical flaws.
One of the concerns is the lack of details and clarity in the methods used and the aims behind using them. For example, all chemicals used (and their characteristics) need to be cited (maybe in a table ) either here or at least in a Sup.Mat.
The authors focused on the 3-octanone although, in the EAG experiment, this compound did not provoke in response at 10 µg/µl (the concentration that was most used in the other experiments). Moreover, why the positive control (hexanal) is not present in the results of the first two experiments (Figures 1 A and B)???
In the preference behavior bioassays the authors used different concentrations for different compounds but their argument for doing so seems very weak!!!
The quality of the figures (fig 1 as an example) is very low and does not allow a good understanding and analysis of the results!!!
Bellow few more points to be considered:
Line 53: define the LmigOr3 at first use.
Line 60: rejection or attraction as stated in the abstract ( line 18)??
Lines 62-63: I think this section might be moved to the ethical statement at the end of the Ms as per the journal's commendations.
Lines 67-86: EAG section, physical conditions ( temperature, lights, the hour of the day..) of the test need to be specified.
Line 85: in the legend of Figure S1, the authors indicate that they used 11 to 12 antennae…please correct one of the statements!!
Lines 87-90: put this section after section 2.2 and maybe the two sections could be joined.
Line 92: indicate which sequence of LmigOR3 was used.
Line 108: what do authors mean with a sample??
Lines 131-132: please indicate how the odor sources were applied
Line 133-135: this section is unclear!!! what n =3-5 indicate??
Line 138: if a locust did not make a choice ..was is it counted for the preference index or discarded?
Line 155: the present Ms does not describe any transcriptome sequencing!!! either include such a section or describe how the RNA was prepared for qPCR!!
Line 186: what do authors mean by the degree of increase...have they calculated some kind of % of increase?
Line 221: details on the two neurons ( A and B) are needed!! how were they identified and separated? what amplitude ranges were used to separate neuron A from neuron B?....
Figure 2 C: Explain what the dashed lines mean.
Line 254-255: this statement needs to be supported by some references...in some cases, concentrations of 1 µg/µl are more likely to be found in natural habitats!!!!
Lines 364-366: This statement is an out of context and needs to be deleted !!!
English if fine.... only moderate editing of English editing is needed.
Author Response
Dear reviewer,
It is my pleased to write you. Thank you very much to have given the valuable suggestions about our manuscript.
We have revised the manuscript and the changes have been highlighted on the basis of your comments. The responses to the your comments have been stated both in website and in this letter according to your suggestion.
The followings are responses to your comments.
Although of some merit, the present Ms lack novelty considering previous similar work with the same topic (see: You et al., 2016; https://doi.org/10.1016/j.ibmb.2016.10.008). Furthermore, the Ms in its present form suffers many critical flaws.
Response: We do appreciate your advice. We have added some sentence to clarify the points which is different from previous topic in Line: 19-22.
One of the concerns is the lack of details and clarity in the methods used and the aims behind using them. For example, all chemicals used (and their characteristics) need to be cited (maybe in a table) either here or at least in a Sup.Mat.
Response: Thanks to your suggestion, the characteristics of all the chemicals used have been added to the supplementary materials (Table S1).
The authors focused on the 3-octanone although, in the EAG experiment, this compound did not provoke in response at 10 µg/µl (the concentration that was most used in the other experiments). Moreover, why the positive control (hexanal) is not present in the results of the first two experiments (Figures 1 A and B)???
Response:
- EAG may not respond to 10 µg/µl because there are some interactions between the neurons which some exhibit excitation, some exhibit inhibition responses to the chemical, but it still caused insect behavior [35]. And we have added the possible explanation about this in the discussion section. In lines: 381-385
- In order to compare the EAG response of antennae to different concentrations of compounds, excluding differences between individuals, antennae were not replaced when stimulated with different concentrations. Before and after each test of the EAG response to 18 compounds, 10% hexanal was used as a positive control to detect whether the isolated antennae was still active. In order to clarify this part more clearly, we have adjusted the structure of Figure 1 and added the supplemental figure S1A, as well as added the relative explanations in the results section. In line: 213-220, 446.
In the preference behavior bioassays the authors used different concentrations for different compounds but their argument for doing so seems very weak!!!
Response: I really appreciate the reviewer's suggestion. The purpose of the behavioral experiment was to test the effect of the compound on the behavior of locusts. Considering that the distance from the odor source to the insect entrance was 49cm (the length of the single arm of the T-maze device), the concentration of the compound transmitted to the insect entrance and perceived by locusts was lower than that of the testing concentration. Furthermore, the EAG reactions were low to all compounds at 1µg/µl. These facts might generate that it cannot detected the compound's real effect on the preference behavior of locusts. To avoid this situation, the 1µg/µl was excluded in behavioral experiment. The concentrations in behavioral bioassay were chosen between 10µg/µl and 100µg/µl. Compared to 100µg/µl, 10 µg/µl was closer to the natural concentration [referred to Chang's preprints literature (bioRxiv preprint, doi: https://doi.org/10.1101/2022.06.21.496967), which have been added in the manuscript]. Therefore, for those had strong EAG reaction at 10µg/µl, 10µg/µl was selected for behavioral bioassay, and 100µg/µl was selected for the rest. And this has been clarified in the manuscript. In lines: 296-305.
The quality of the figures (fig 1 as an example) is very low and does not allow a good understanding and analysis of the results!!!
Response: I'm obliged for your suggestion. We have improved our figures into high quality.
Bellow few more points to be considered:
Line 53: define the LmigOr3 at first use.
Response: Line 68: Thanks for your suggestion. LmigOr3 was defined when first used.
Line 60: rejection or attraction as stated in the abstract ( line 18)??
Response: Line 19 and 78: Locusts tend to reject 3-octanone, and LmigOR3 mediates this rejection behavior. After RNAi reduced the expression of LmigOR3, the locust's rejection of 3-octanone disappeared and became attracted. Therefore, it is said in the abstract that RNAi of LmigOR3 increased attraction by 3-octanone to locusts.
Lines 62-63: I think this section might be moved to the ethical statement at the end of the Ms as per the journal's commendations.
Response: Thanks for your valuable suggestion. Lines 484-485: The statement has been moved to the end.
Lines 67-86: EAG section, physical conditions (temperature, lights, the hour of the day..) of the test need to be specified.
Response: Thank you for valuable suggestion. Lines 91-92: Physical conditions of EAG experiments have been added.
Line 85: in the legend of Figure S1, the authors indicate that they used 11 to 12 antennae…please correct one of the statements!!
Response: My apologies for our carelessness. Line 106: One of these statements has been corrected.
Lines 87-90: put this section after section 2.2 and maybe the two sections could be joined.
Response: I do appreciate your advice. Lines 109-112: The two sections have been merged.
Line 92: indicate which sequence of LmigOR3 was used.
Response: I do appreciate your advice. Line 115: The name of the sequence used has been indicated.
Line 108: what do authors mean with a sample??
Response: Line 132-133: In the WM-FISH experiment, one antenna was made into one sample slide. The number of samples was 3, which means that the presence of the same signal was observed in 3 different slides (three antenna). And this has been clarified in the manuscript.
Lines 131-132: please indicate how the odor sources were applied
Response: Thank you for your suggestion. Lines 155-158: The method of application of odor source has been indicated.
Line 133-135: this section is unclear!!! what n =3-5 indicate??
Response: Lines 133-135: It has been modified. n =3-5 indicates the number of replicates. In lines: 159-161.
Line 138: if a locust did not make a choice .. was is it counted for the preference index or discarded?
Response: Line 164-165: If the locust did not make a choice, it was discarded. And this has been clarified in the manuscript.
Line 155: the present Ms does not describe any transcriptome sequencing!!! either include such a section or describe how the RNA was prepared for qPCR!!
Response: Thank you for the suggestion. Line 182-186: The description of the relevant steps has been added.
Line 186: what do authors mean by the degree of increase...have they calculated some kind of % of increase?
Response: Line 227: The degree of increase here refers to ∆EAG, which means that the EAG reaction of these compounds was stronger than that of other compounds. The "degree of increase in reaction" has been amended to "EAG response ".The total increase percentage of EAG has been added in the Figure S1B. In lines 241-245.
Line 221: details on the two neurons ( A and B) are needed!! how were they identified and separated? what amplitude ranges were used to separate neuron A from neuron B?....
Response: Thank you for your valuable suggestion. Using the Auto spike software to identify spikes amplitudes, it was found that the amplitudes could be separated into two parts. According to the reference, they are divided into two types of neurons, those with large amplitudes are named as A neurons, and those with small amplitudes are named as B neurons [28]. The amplitude size in SSR experiment was affected by individual and experimental operation, and the absolute value is not fixed, only from whether it can be separated into two parts to judge the number of neurons. But the relative size of amplitude is fixed, and the large amplitude is always the A neuron. The amplitude difference between neurons A and B ranges from 3 to 21mv.
Figure 2 C: Explain what the dashed lines mean.
Response: Thank you for your valuable suggestion. Line 292-293: Figure 2 C: Relevant explanations have been added.
Line 254-255: this statement needs to be supported by some references...in some cases, concentrations of 1 µg/µl are more likely to be found in natural habitats!!!!
Response: I appreciate your valuable suggestion. We have added the relevant literature and made it more clearly in the results. Because the distance from the source of the odor to the insect entrance (about 49 cm) weakened the odor concentration practically perceived by locusts, and the aim of the experiment was to test the effect of the compound on locust behavior, furthermore, the low EAG response at 1µg/µl, so we excluded 1µg/µl here and only compared 10µg/µl and 100µg/µl. And according to the literature, we found that 10µg/µl was closer to the concentration in the natural environment than 100µg/µl [21]. In lines 296-305.
Lines 364-366: This statement is an out of context and needs to be deleted !!!
Response: Thank you for your valuable suggestion. Lines 364-366: This statement has been deleted.

Reviewer 2 Report
Reviewer Comments #: -
The manuscript describes “3-octanone, detected by the locust odorant receptor LmigOR3, induces locusts to keep away.”. The author tries to figure out the role of 3-octanone using known techniques like EAG, SSR and RNAi. Techniques used in the manuscript are up to date and the experiments are well performed. Overall, the manuscript is well organized with interesting functional information. However, I found certain things which need clarification to support and strengthen the conclusion.
Title: Please make the title simple.
Abstract: Please rearrange it because few sentences are not clear especially the second sentence.
Update the introduction with latest references which are missing. In addition to it please extend it.
Why do you select 4th instar not the last instar? Can you incorporate the details.
“The antennae were removed from the base, and the tip of the antenna was removed”. What do you mean by this. When you cut the antenna from base, how can you judge the signals are correct?
How do you standardize the given concentrations for EAG. Do you have any reference or please provide the details.
How many references gene you have checked. Please provide the details. It’s recommended to check minimum two reference genes.
If possible, please provide the graphical abstract.
In the overall structure of your publication the discussion part is weakly written please focus on it and extend it with comparison to the current findings and more relevant references.
I definitely recommended this manuscript should pass through English correction with a expertise native speaker or through a company.
Conclusive remarks:
The manuscript contains interesting and significant findings. It still needs some major corrections (Typo’s, italics, and sentence repetitions). I still ask the authors to rearrange and extend information on certain parts of the manuscript especially about the discussion with additional references. I do think that the manuscript contains important issues, information, interesting approaches and techniques, which can lead to proper understanding the role of 3-octanone and its functional mechanism in detail. So, I consider this manuscript suitable for the publication after the suggested clarifications in agriculture.
I definitely recommended this manuscript should pass through English correction with a expertise native speaker or through a company.
Author Response
Dear reviewer,
It is my pleased to write you. Thank you very much to have given the valuable suggestions about our manuscript.
We have revised the manuscript and the changes have been highlighted on the basis of your comments. The responses to your comments have been stated both in website and in this letter according to your suggestion.
The followings are responses to your comments.
Review 2:
Reviewer Comments #: -
The manuscript describes “3-octanone, detected by the locust odorant receptor LmigOR3, induces locusts to keep away.”. The author tries to figure out the role of 3-octanone using known techniques like EAG, SSR and RNAi. Techniques used in the manuscript are up to date and the experiments are well performed. Overall, the manuscript is well organized with interesting functional information. However, I found certain things which need clarification to support and strengthen the conclusion.
Title: Please make the title simple.
Response: Line 1-4: Thank you for your valuable suggestion. The title has been improved.
Abstract: Please rearrange it because few sentences are not clear especially the second sentence.
Response: Thank you for your valuable suggestion. Line 9-12: The abstract has been reorganized.
Update the introduction with latest references which are missing. In addition to it please extend it.
Response: Many thanks for your valuable suggestion, the introduction has been extended and references that can be updated have been updated. In lines: 492, 502-516, 521-526.
Why do you select 4th instar not the last instar? Can you incorporate the details.
Response: the 4th instar locusts were easier to operate and sensitive enough to odorants [3].
“The antennae were removed from the base, and the tip of the antenna was removed”. What do you mean by this. When you cut the antenna from base, how can you judge the signals are correct?
Response: This means that the epidermis at both ends of the antenna is destroyed, so that the two electrodes and the antenna can form a circuit path. According to the published paper [32], hexanal was used as a positive control to confirm that normal electrophysiological responses of the antenna to be evoked from start experiment to the end, in other word, the cut off antenna can keep active sustainably.
How do you standardize the given concentrations for EAG. Do you have any reference or please provide the details.
Response: We have added the statement about standardization of concentrations of odors for EAG. In lines: 94-95.
How many references gene you have checked. Please provide the details. It’s recommended to check minimum two reference genes.
Response: Thanks for your valuable suggestion. In qPCR, one reference gene (LmigActin) was used, and the primer information has been added to the supplemental material. In semi-quantitative PCR, two reference genes (LmigActin and LmigOR2) were detected.

If possible, please provide the graphical abstract.
Response: Thanks for your valuable suggestion. The graphical abstract has been provided.
In the overall structure of your publication the discussion part is weakly written please focus on it and extend it with comparison to the current findings and more relevant references.
Response: Thanks for your valuable suggestion. Lines 352-356, 360-366, 381-385, 396-404, 411-415, 430-438: The content of discussion section has been added.
I definitely recommended this manuscript should pass through English correction with a expertise native speaker or through a company.
Response: Thanks a lot. This manuscript has been edited by native language experts.
Conclusive remarks:
The manuscript contains interesting and significant findings. It still needs some major corrections (Typo’s, italics, and sentence repetitions). I still ask the authors to rearrange and extend information on certain parts of the manuscript especially about the discussion with additional references. I do think that the manuscript contains important issues, information, interesting approaches and techniques, which can lead to proper understanding the role of 3-octanone and its functional mechanism in detail. So, I consider this manuscript suitable for the publication after the suggested clarifications in agriculture.
Best regards
Yours,
Long Zhang

Reviewer 3 Report
s a reviewer, I have assessed the provided article titled "3-octanone, detected by the locust odorant receptor LmigOR3, induces locusts to keep away" for the journal Agriculture (ISSN 2077-0472). Overall, the article presents valuable research on the avoidance behavior of locusts in response to specific chemicals. However, there are a few areas that require revision and improvement in terms of scientific language, grammar, and overall clarity.
The abstract provides a concise summary of the study, but it could benefit from further clarification. In particular, the objectives and significance of the research should be explicitly stated. Additionally, the abstract should mention the methodology employed to investigate the behavioral and electrophysiological responses of locusts.
In the introduction, it would be helpful to provide more background information on the Locusta migratoria species and its economic impact as an agricultural pest. Moreover, the rationale for studying the avoidance of harmful substances in locusts could be expanded upon, along with the potential implications of this research for pest management strategies.
The methods section requires more detail to ensure reproducibility. Specifically, the experimental design, the number of replicates, and the statistical analyses performed should be clearly described. Furthermore, the materials used for the electrophysiological and behavioral experiments should be specified.
The results section presents the main findings of the study, but the presentation could be improved for better clarity. It would be beneficial to provide a more structured and organized presentation of the results, possibly using tables or figures to present the data more effectively. Additionally, the authors should include the concentration of the chemicals tested and the duration of the behavioral assays.
The discussion section should focus on interpreting the results in the context of the broader literature. The authors should compare their findings with previous studies, discussing similarities and differences. Furthermore, the implications and potential applications of the results should be addressed, emphasizing the contribution of the study to the field of insect olfactory recognition.
Finally, the article would benefit from a thorough proofreading to address grammatical errors and improve the overall scientific language. Ensuring clarity and coherence of the manuscript will enhance its readability and scientific quality.
In conclusion, the article presents valuable research on the avoidance behavior of locusts in response to specific chemicals. However, it requires some revisions to improve the scientific language, grammar, and clarity. Additionally, the article would benefit from providing more details on the experimental methods, presenting the results more effectively, and strengthening the discussion section by incorporating relevant literature.
s a reviewer, I have assessed the provided article titled "3-octanone, detected by the locust odorant receptor LmigOR3, induces locusts to keep away" for the journal Agriculture (ISSN 2077-0472). Overall, the article presents valuable research on the avoidance behavior of locusts in response to specific chemicals. However, there are a few areas that require revision and improvement in terms of scientific language, grammar, and overall clarity.
The abstract provides a concise summary of the study, but it could benefit from further clarification. In particular, the objectives and significance of the research should be explicitly stated. Additionally, the abstract should mention the methodology employed to investigate the behavioral and electrophysiological responses of locusts.
In the introduction, it would be helpful to provide more background information on the Locusta migratoria species and its economic impact as an agricultural pest. Moreover, the rationale for studying the avoidance of harmful substances in locusts could be expanded upon, along with the potential implications of this research for pest management strategies.
The methods section requires more detail to ensure reproducibility. Specifically, the experimental design, the number of replicates, and the statistical analyses performed should be clearly described. Furthermore, the materials used for the electrophysiological and behavioral experiments should be specified.
The results section presents the main findings of the study, but the presentation could be improved for better clarity. It would be beneficial to provide a more structured and organized presentation of the results, possibly using tables or figures to present the data more effectively. Additionally, the authors should include the concentration of the chemicals tested and the duration of the behavioral assays.
The discussion section should focus on interpreting the results in the context of the broader literature. The authors should compare their findings with previous studies, discussing similarities and differences. Furthermore, the implications and potential applications of the results should be addressed, emphasizing the contribution of the study to the field of insect olfactory recognition.
Finally, the article would benefit from a thorough proofreading to address grammatical errors and improve the overall scientific language. Ensuring clarity and coherence of the manuscript will enhance its readability and scientific quality.
In conclusion, the article presents valuable research on the avoidance behavior of locusts in response to specific chemicals. However, it requires some revisions to improve the scientific language, grammar, and clarity. Additionally, the article would benefit from providing more details on the experimental methods, presenting the results more effectively, and strengthening the discussion section by incorporating relevant literature.
Author Response
Dear reviewer,
It is my pleased to write you. Thank you very much to have given the valuable suggestions about our manuscript.
We have revised the manuscript and the changes have been highlighted on the basis of your comments. The responses to your comments have been stated both in website and in this letter according to your suggestion.
The followings are responses to your comments.
Review 3:
As a reviewer, I have assessed the provided article titled "3-octanone, detected by the locust odorant receptor LmigOR3, induces locusts to keep away" for the journal Agriculture (ISSN 2077-0472). Overall, the article presents valuable research on the avoidance behavior of locusts in response to specific chemicals. However, there are a few areas that require revision and improvement in terms of scientific language, grammar, and overall clarity.
The abstract provides a concise summary of the study, but it could benefit from further clarification. In particular, the objectives and significance of the research should be explicitly stated. Additionally, the abstract should mention the methodology employed to investigate the behavioral and electrophysiological responses of locusts.
Response: Many thanks for your valuable suggestion. Lines 10-12: Methods for investigating locust behavior and electrophysiological responses have been added to the abstract.
In the introduction, it would be helpful to provide more background information on the Locusta migratoria species and its economic impact as an agricultural pest. Moreover, the rationale for studying the avoidance of harmful substances in locusts could be expanded upon, along with the potential implications of this research for pest management strategies.
Response: I sincerely appreciate your valuable suggestion. Some information has been added in the introduction. In lines: 30-34, 39-51.
The methods section requires more detail to ensure reproducibility. Specifically, the experimental design, the number of replicates, and the statistical analyses performed should be clearly described. Furthermore, the materials used for the electrophysiological and behavioral experiments should be specified.
Response: I sincerely appreciate your valuable suggestion. Some methods in details have been added. In lines: 81-86, 91, 155-161, 182-186, 199-206.
The results section presents the main findings of the study, but the presentation could be improved for better clarity. It would be beneficial to provide a more structured and organized presentation of the results, possibly using tables or figures to present the data more effectively. Additionally, the authors should include the concentration of the chemicals tested and the duration of the behavioral assays.
Response: Thank you for your valuable suggestion. Lines 213-220, 241-245, 308, 336-339: The results section has been modified. And the concentration of the chemicals tested and the duration of the behavioral assays have been documented in the relative method.
The discussion section should focus on interpreting the results in the context of the broader literature. The authors should compare their findings with previous studies, discussing similarities and differences. Furthermore, the implications and potential applications of the results should be addressed, emphasizing the contribution of the study to the field of insect olfactory recognition.
Response: I'm truly obliged for your valuable suggestion. According to the suggestions of reviewer, the content of discussion section has been added. In lines: 352-356, 360-366, 381-385, 396-404, 411-415, 430-438.
Finally, the article would benefit from a thorough proofreading to address grammatical errors and improve the overall scientific language. Ensuring clarity and coherence of the manuscript will enhance its readability and scientific quality.
Response: This manuscript has been edited by native language experts.
In conclusion, the article presents valuable research on the avoidance behavior of locusts in response to specific chemicals. However, it requires some revisions to improve the scientific language, grammar, and clarity. Additionally, the article would benefit from providing more details on the experimental methods, presenting the results more effectively, and strengthening the discussion section by incorporating relevant literature.
Best regards
Yours,
Long Zhang

Round 2
Reviewer 2 Report
Reviewer #:
Now, this is a very well-conceived and written paper.
The author incorporated the particulars in the present revised version of the manuscript.
Please revise the incorporation perfectly. I still found some typos which needs correction.
Once everything has been included, carefully review the references to ensure that they are all in accordance with the journal's format.
I agree that this article should be published in Agriculture after such adjustments.